# Review of Harmful Algal Blooms (HABs) Causing Marine Fish Kills: Toxicity and Mitigation

**DOI:** 10.3390/plants12233936

**Published:** 2023-11-22

**Authors:** Jae-Wook Oh, Suraj Shiv Charan Pushparaj, Manikandan Muthu, Judy Gopal

**Affiliations:** 1Department of Stem Cell and Regenerative Biotechnology, Konkuk University, Seoul 143-701, Republic of Korea; ohjw@konkuk.ac.kr; 2Department of Research and Innovation, Saveetha School of Engineering, Saveetha Institute of Medical and Technical Sciences (SIMATS), Thandalam, Chennai 602105, Tamil Nadu, India; bhagatmani@gmail.com

**Keywords:** harmful algal blooms, marine, fish kills, environment, mitigation, water pollution

## Abstract

Extensive growth of microscopic algae and cyanobacteria results in harmful algal blooms (HABs) in marine, brackish, and freshwater environments. HABs can harm humans and animals through their toxicity or by producing ecological conditions such as oxygen depletion, which can kill fish and other economically or ecologically important organisms. This review summarizes the reports on various HABs that are able to bring about marine fish kills. The predominant HABs, their toxins, and their effects on fishes spread across various parts of the globe are discussed. The mechanism of HAB-driven fish kills is discussed based on the available reports, and existing mitigation methods are presented. Lapses in the large-scale implementation of mitigation methods demonstrated under laboratory conditions are projected. Clay-related technologies and nano-sorption-based nanotechnologies, although proven to make significant contributions, have not been put to use in real-world conditions. The gaps in the technology transfer of the accomplished mitigation prototypes are highlighted. Further uses of remote sensing and machine learning state-of-the-art techniques for the detection and identification of HABs are recommended.

## 1. Introduction

Harmful algal blooms (HABs) occur when toxin-producing algae grow excessively in a water body. Depending on the water body, they can be classified as marine or freshwater HABs. HABs have been reported since the 1500s [1,2,3,4,5,6,7,8]. Excessive algal growth, or algal bloom, can be green, blue–green, red, or brown, depending on the type of algae. HABs result from the rapid proliferation of cyanobacteria, marine microalgae, diatoms, dinoflagellates, and raphidophytes, endangering people, animals, or the local ecology. To date, around 300 hundred species of microalgae occasionally trigger massive events known as blooms, of which about 75 have been known to generate toxins (https://hab.ioc-unesco.org/what-are-harmful-algae/, accessed on 28 September 2023).

The influence of climate change effects such as rising sea levels, saltier freshwater, nutrient pollution, and ocean acidification are factors that are correlated with HAB spikes [9,10,11]. These conditions enable the invasion of HABs, thereby compromising ecological integrity and wreaking havoc on marine communities and the local economy of people who depend on them [12,13]. The economic and public health impacts of HABs can be manifold. Millions of dollars have been invested annually to address the known HAB-related impacts on public health, commercial fisheries, recreation, tourism, environmental monitoring, and bloom management. Public health impacts account for the largest economic impacts, followed by commercial fisheries and tourism. Even a single HAB can be extremely costly. The corresponding issues, such as impacts on secondary industries (e.g., aquaculture suppliers) and decline in consumer confidence (e.g., failure to purchase seafood in restaurants or reserve fishing charter trips), are unassessed. The environmental impacts can be huge, with HABs affecting marine fauna and flora as well as human health and welfare. The dinoflagellates have the greatest number of harmful species, some of which produce potent toxins. The dinoflagellate toxins directly affect the ecosystem and are suspected to cause mass fish kills [14,15]. Moreover, it has been reported that toxins from dinoflagellates bio magnify up the food chain and cause mass mortality of finfish [16]. The issue of toxic dinoflagellate algal blooms has been more frequent in recent times [11,17]. In humans, toxicity results from the ingestion of contaminated seafood products (fish or shellfish), skin contact with toxin-contaminated water, or the inhalation of aerosolized toxins or noxious compounds. In the case of food-borne poisonings, HAB toxins are bio-concentrated, most often without harming the vector marine organism; this is then transferred up through the food web to humans [18]. In addition to the direct impacts on human health, these toxic outbreaks have associated consequences on other components of human well-being in terms of their socio-economic impacts, as well as associated costs.

Each year, HABs are responsible for thousands of incidents all over the globe, with symptoms that include vomiting, diarrhea, dizziness, or, in extreme cases, even death, as well as respiratory issues in people who breathe in toxic aerosols. HAB outbreaks can have a big impact on local economies and lead to the closure of fisheries, aquaculture, and recreational areas, the loss of fishery products, with subsequent declines in businesses, tourism, and associated services [19]. In fact, the National Centres for Coastal Ocean Science proposed an economic loss of USD 10–100 million per year resulting from HAB events in the USA. In the temperate latitudes of South Africa, Europe, Asia, North America, Australia, and South America, HABs have been reported to cause amnesic (ASP), azaspiracid (AZP), diarrhetic (DSP), neurotoxic (NSP), and paralytic (PSP) shellfish poisonings [20]. Algal blooms can also impact aquatic animals indirectly. When algal cells die and sink to the bottom, they provide a rich food source for bacteria, which, during decomposition, consume dissolved oxygen. This is a major cause of the low-oxygen dead zones that plague coastal waters worldwide. According to scientists, there was roughly a three-fold increase in harmful algal blooms reported from 2000 to 2020 (https://www.thehindu.com/news/cities/Kochi/rise-in-harmful-algal-blooms-in-arabian-sea-posing-health-risk-to-fish-consumers-say-scientists/article66536326.ece, accessed on 21 February 2023). Figure 1 gives an overview of the various categories of HABs that are involved in marine fish kills. As shown in the figure, the associated impacts of HABs have been listed.

Understanding the means of fish kills can contribute to mitigation strategies to reduce the risks resulting from HAB events. The current mitigation methods being practiced fall within broad categories of physical, chemical, and biological methods. HAB toxicity has been elaborately dealt with on individual or population levels as well as in cellular and ecosystem studies. Additionally, the mechanisms for those toxic effects, such as hypoxia, reactive oxygen species (ROS), and toxins (shellfish toxins, haemotoxins, or cytotoxins), have been previously described.

In the following survey, we present a comprehensive overview of up-to-date reports on HABs affecting marine fish, resulting in fish kills. The predominant HAB species involved and the potential toxins that mediate fish kills are discussed. The mechanism behind HAB toxicity leading to fish kills is presented. The need to expand and fill in the gaps in existing knowledge is suggested. Mitigation measures and lapses in the implementation of mitigation methods are presented.

## 2. Algal Bloom-Associated Toxicity

HABs, as their name implies, are able to trigger a cascade of harmful effects on marine ecology, marine fauna and flora, and human welfare. Table 1 summarizes the HAB events that have occurred in the recent past (2018 onwards up to date), mapping the location and the time duration of each HAB event.

HABs restrict light penetration, which is potentially fatal to macrophytes such as water lilies. In addition, HABs consume dissolved oxygen in the water, triggering adverse metabolic reactions in fish. The situation is further aggravated when the algae die, leading to the consumption of more dissolved oxygen. Such oxygen-deficient water bodies are known as “dead-zones” [10]. It is to be noted that dead zones across the coastal and open oceans have increased ten-fold since 1950 (https://www.weforum.org/agenda/2018/01/dead-zones-in-our-oceans-have-increased-dramatically-since-1950-and-we-re-to-blame/, accessed on 18 January 2023). The recurrence of dead zones in large water bodies such as Lake Erie, Chesapeake Bay, and the Gulf of Mexico, poses challenges that call for immediate attention [10,21]. In most cases, dead zone formation has been associated with the growth of toxic, dense algal blooms. Further, the increased rate of photosynthesis due to eutrophication limits the availability of inorganic carbon and increases the alkalinity of the water [22]. Sudden changes in pH could result in loss of eye sight in aquatic organisms, affecting their survivability [23]. The existence of dead zones has been found in over 400 marine systems, particularly at the zone where nutrient-rich river water meets the sea/ocean, i.e., near estuaries, lagoons, and coastal areas, such as the juncture of the Mississippi River and the Gulf of Mexico [24]. Thus, the presence of algal blooms causes major hindrances to economies that depend on water resources such as commercial fishing and recreational activities. The case for HABs is even worse, as the toxins have the potential to cause either morbidity or mortality in living beings [25].

Toxins such as domoic acid, saxitoxin, brevetoxin, okadaic acid, and ciguatoxin trigger many illnesses such as Ciguatera Fish Poisoning (CFP), Amnesic Shellfish Poisoning (ASP), Diarrheic Shellfish Poisoning (DSP), and Paralytic shellfish poisoning (PSP) [26,27,28]. Thus, addressing and understanding algal blooms have become an objective of the United Nations (UN) Sustainable Development Goals (SDG), primarily Goal 6 (clean water and sanitation) and Goal 14 (life below water).

In some instances, toxins are not actively excreted; the toxin is released when a HAB ruptures or is eaten by another creature [29]. Some HABs generate one specific type of toxin only, whereas others produce several. Skin contact, ingesting infected seafood, swallowing water while swimming, and, in the case of animals, lapping the contaminated water or algae off their fur after swimming through an algal bloom are the most typical ways that the HAB toxins enter the animal/human systems. Some people experience immediate allergic reactions; direct exposures can irritate the skin, nose, throat, and eyes, as well as cause respiratory system inflammation [30]. That includes the effects of ingestion, including nausea, vomiting, and diarrhea. HAB toxins have been connected to neurological issues and liver illness in humans [30,31]. It has been confirmed that marine blooms not only pose a serious risk to public health but also reduce the efficiency of zooplankton (water supply energy efficiency) to consume algae, which can indirectly restrict algal growth [27,32]. All these have huge repercussions in affecting the socio-economic conditions of local people and their regional economy.

## 3. Surveying Specific Reports of HAB Fish Kills

Fish deaths have been linked to a variety of toxic algae, although in many cases, it is difficult to identify the specific toxins or poisoning mechanism [27]. Nevertheless, several toxin types, many of which are ichthyotoxic, have been discovered in several types of algae. Disorientation, loss of balance, and occasional hyperactivity are some of the clinical indications of algal neurotoxicity. Hypoxia is brought on by other algae that mechanically block or harm the gills [33,34]. The global algal bloom status report noted a rise in the incidence, severity, and geographic spread of HABs in estuarine and fresh waters [35]. Consequently, HABs have been linked with both wild and farmed fish deaths. In the sections that follow, we will feature the predominant HAB species and their effects on fish based on the reports published thus far.

### 3.1. Prymnesium Parvum

Significant HABs that compromise the stability of aquatic ecosystems can be caused by *Prymnesium parvum* [36]. This mixotrophic haptophyte has been reported to orchestrate significant ecological effects, including enormous fish fatalities, in many environments because of its ichthyotoxicity. This organism is said to be mostly of marine origin but also tends to live in inland water with high mineral content [37]. *P. parvum*, sometimes known as “golden alga”, has been reported to cause huge fish kills. For instance, they have been linked to approximately 135 tons of farmed Atlantic salmon (*Salmo salar*) in Norway that were killed during a bloom in 2007 [38]. The chemical composition of the toxins of *P. parvum*, collectively referred to as prymnesins, has been found to contain biomolecules like glycolipids, galactolipids, proteolipids, and lipid–carbohydrate molecules [36], and the way they elicit toxicity are being investigated. Clinton et al. examined the gill transcriptome sensitivity of juvenile rainbow trout as a function of toxin dosage and fish phenotype [39]. Typically, the fish were exposed to prymnesins for 4–5 h. The study demonstrated the activation of acute pro-inflammatory cytokines that cause severe gill malfunction after exposure to the toxins, and these responses are independent of toxin dosage and fish phenotype. *P. parvum* excretes toxins to catch prey; however, it is hypothesized that *Prymnesium* toxins are released again after cell-to-cell contact in a process known as “toxin-mediated micropredation” [39,40]. In fact, reports of direct physical contact with animals have been reported [41]. That could explain why finfish are more prone to being injured by toxins compared to shelled species like prawns [36]. According to Qin et al., other parameters, such as water temperature, salinity, and nutritional status, particularly the levels of N and P, play a role in the cytotoxicity effect of *P. parvum* toxins [36]. Prymnesins have hemolytic and cytotoxic qualities that may affect the integrity of cell membranes, resulting in an impairment in cell permeability [42]. Though there is no consensus on the number of cells that could be lethal to fish, a dosage of 10,000–20,000 cells/mL has been reported to cause fish kill [43]. The research to disclose other secondary metabolites responsible for *P. parvum*’s toxicity is still in progress [44]. Many species of *Prymnesium* are recognized to produce toxic compounds that impact erythrocytes and gill-respiring organisms [45]. *P. parvum* causes periodical HABs, whose cell densities grow quickly and are able to release powerful ichthyotoxins [44].

Local fauna may be significantly threatened by persistent Prymnesium HABs, which impart enormous damage to the ecosystem and the local economy [46]. A majority of investigations have linked *P. parvum* and *P. parvum f. patelliferum* (belonging to the same species), which are alternate phases in a haploid–diploid life cycle, owing to the extent of their spread and the frequency of their toxic occurrences [46,47]. The reported case of *P. parvum* and its bloom’s connection with sick and dead fish was documented in 1938 around the brackish waters of Northern Europe [48]. Since then, the algae’s connection to seasonal toxic blooms and large fatalities in fish ponds and in native gill-respiring animals has been well-documented [49].

### 3.2. Karenia Mikimotoi

*Karenia mikimotoi* is a dinoflagellate species from the genus *Karenia*. It was first reported in Japan in 1935, and since then, it has appeared in other parts of the world, such as the east coast of the United States, Norway, and the English Channel [50]. *K. mikimotoi* has yellow–brown chloroplasts and, like other species in its genus, is able to activate photosynthetically [51]. HABs of *K. mikimotoi* have been affecting large numbers of marine species around the world for more than 80 years. Large-scale fish kills have been documented along the coastal seas of Europe and Asia, resulting in significant economic losses [50,52]. Three main mechanisms by which *K. mikimotoi* may cause fish deaths include [53]: (1) the production of reactive oxygen species (ROS) that damage fish respiratory systems and destabilize their antioxidant defenses; (2) fish suffocation due to lack of dissolved oxygen; and (3) the release of cytotoxic toxins like gymnocins, hemolysins, and specific polyunsaturated fatty acids. Another evolving neurotoxin known as gymnodimines has also been reported in fish kills and is produced by *Karenia selliformis* [54].

To identify differentially expressed proteins, entire proteomes of medaka (the Japanese rice fish) were studied using 2-DE (two-dimensional electrophoresis) by varying the toxin dosage of *K. mikimotoi* (LT25, LT50, and LT90) [53]. A total of 35 differential protein locations with at least two-fold variations were confirmed with mass spectrometric analysis. Typically, these regions cover some 19 proteins that are correlated with strong inflammatory and oxidative stress responses. Several unfavorable symptoms that emerged during the exposure time, such as asphyxiation, losing balance, and body jerking, which are all symptoms of muscle injury, were observed. Kwok et al. [53] reported a novel approach employing molecular analysis to study ichthyotoxicity processes; the exact toxicity pathways of fish kills are still at the infant stage, as inferred by these authors.

Previous studies have established the hemolytic and cytotoxic nature of *K. mikimotoi*, as well as its lethality and the cytotoxic mechanisms operating against marine organisms [50]. *K. mikimotoi* has had numerous names over its lengthy and intricate categorization history. Though first identified as *Gymnodinium mikimotoi*, it was finally named by Dr. Shoichi Miyake and Konan Kiyo from sample waters of Gokasho Bay, Honshu, Japan [50]. A snapshot of the various records of this dinoflagellate bloom is presented in detail by Li, Xiaodong, et al. [50], leading to mortalities of Pacific abalone (*Haliotis discus hannai*), Japanese common squid (*Todarodes pacificus*) and chum salmon (*Oncorhynchus keta*) [55], trout, cod, and eels [56,57], Pacific salmon, and Atlantic salmon [50,58]. Ironically, these blooms in Chile were reported to have occurred only once, whereas, in the case of countries like China, these blooms have recurred periodically since 2002, causing havoc to the ecological balance and local economy [59]. Valuable marine edible fish along the East China Sea and its related areas have been documented to be vastly affected. In fact, an economic loss of USD 50.8 million was reported in the area around Hong Kong [58]. With over 100 blooms documented from 2006 to 2018, *K. mikimotoi* blooms are now a frequent and regular yearly environmental calamity in China [59,60]. It is predicted that blooms will continue to be often observed along China’s coastlines for a very long time.

According to the majority of field reports and laboratory testing, *K. mikimotoi* can be poisonous to fish, particularly harming fish gills. Prior to death, the fish exhibit abnormal behavioral activities and elevated opercular rates [50]. In these dead fish, gill abnormalities, excessive mucus secretion, and knotted filaments were perceived to cause fish suffocation [61]. Salmon appeared agitated and stuck to the surface before they sank down as soon as the blooms reached their tanks [62]. Sloughing and oedematous epithelia were observed in the dead fish, along with excess mucus secretion and gill blemishes [50]. In fact, in a pilot test, the deceased fish’s gills did not exhibit mechanical obstruction by the alga, but they did exhibit histological alterations, such as necrotic atrophy and breakdown of the lamellar epithelium [63,64]. Reports confirmed a rheological stickiness of water—caused by both the algae and the fish mucus, which may play a role in the cytotoxicity of *K. mikimotoi* [65]. Severe gill damage was noted in farmed fish amidst algal blooms in Hong Kong waters [66]. Clinical examination of salmon livers by Mitchell and Rodger in 2007 revealed widespread coagulative necrosis in addition to tissue destruction in the gills [67]. Even in waters with high dissolved oxygen (DO), Li et al. (2017) [60] observed that *K. mikimotoi* may be fatal to turbot (*Scophthalmus maximus*) and that the death rate rose when the DO content was not regulated. However, no additional harmful consequences have been observed besides mortality, behavioral issues, and gill damage. It is yet unknown if *K. mikimotoi* has sublethal effects on fish, such as immunotoxicity effects or developmental toxicity.

### 3.3. Karenia Brevis

Florida’s west coast experiences blooms of the toxic alga *Karenia brevis*, sometimes infamously known as “Florida red tides”, every year, which results in significant fish mortality. However, there is little quantitative data available on the ecological consequences of *K. brevis* on finfish communities. Stumpf et al. [68] evaluated the intensity of *K. brevis* blooms across time from 1953 to 2019 over the shorelines of southwest Florida and developed a Bloom Severity Index (BSI). In their study, they demonstrated that the effects of blooms on the respiratory system are not uniform in time and space. While some areas may experience effects for several months in a row during major blooms, the entire coast does not experience effects continuously from start to finish due to other factors, including the strength of offshore/onshore winds. *K. brevis* brevetoxins (PbTx) have been detected in the tissues of the dead fish [69].

Gannon et al. [70] investigated variations in fish densities and fish diversities in connection with *K. brevis* in five habitats around Sarasota Bay (close to the Gulf of Mexico) during the 2004 to 2007 summers. Physical parameters such as salinity, water temperature, and turbidity were recorded. Red tides resulted in considerably decreased species richness and fish density in terms of catch per unit effort (CPUE) across all environments. In 4 out of 5 habitats, Shannon–Wiener diversity indices (which reflect species evenness and richness) were considerably lower during red tides. This was supported by a regression tree and canonical correspondence analysis, which revealed that the *K. brevis* density had a significant influence on the fish population and distribution. In fact, Florida’s database on fish kills relates to 96% of fish mortality from the 2003 to 2007 red tides [70]. It is inferred that these brevetoxins have the potential to alter the ecosystem by impacting the fish larvae survival rate, altering phytoplankton composition, and varying the population within an ecosystem [71]. However, it is inconclusive whether the change in community structure and fish density is either through mass mortality (of fish) or mass emigration. The episode of massive fish deaths and high PbTx concentrations in fish tissues, however, imply that brevetoxin-induced mortality was a key contributor, leading to alterations at the community level [72]. Brevetoxins, which are fatal to fish, have lately been shown to accumulate in tissues of higher trophic animals like birds, bottlenose dolphins, manatees, and humans [72,73,74,75]. Brevetoxins may be identified in fish tissues up to a year after red tides [76], and they were discovered in Sarasota Bay fish tissues many months after these specific red-tide occurrences, showing that they persist in the food chain for a long time. From the Florida red tide observations, it can be hypothesized that, besides mass mortality, brevetoxins modify the fish abundance, species diversity, and community diversity (population) of local habitats.

### 3.4. Heterosigma Akashiwo

Over the past few decades, the frequency of HABs of *Heterosigma akashiwo* (Hada) along coastal areas has increased. *H. akashiwo* is a species of microscopic algae of the class Raphidophyceae [77], which is a marine alga that leads to HABs. The species name akashiwo is from the Japanese for “red tide”; other synonyms include *Olisthodiscus luteus* [78] and *Entomosigma akashiwo* [79]. *H. akashiwo* recently caused a fish-killing bloom event in 2021 that sparked a scientific investigation, but the ichthyotoxic pathway and environmental factors that support its proliferation are yet unknown [80]. According to the RT gill-W1 bioassay study, the *H. akashiwo* strain is only cytotoxic at higher cell concentrations (>47,000 cells mL^−1^) and after cell breakage. It was proposed that high levels of long-chain PUFA synthesis from high cell densities caused salmon mortality [80].

The devastating consequences of these algae toxins on fish and other species have been a high concern [81]. *H. akashiwo* blooms have been reported throughout major continents, from the USA–Canada, Bermuda, Europe, Asia, and New Zealand [81]. In a study, this bloom was reported to cause losses on the scale of USD 4–5 million dollars [82]. It has been a mystery how *H. akashiwo* and other similarly related raphidophytes, such as *Chattonella* spp. and *Fibrocapsa* spp., kill fish [83]. However, compared to the other HABs, in the case of this genera, some knowledge has been gathered about the potential toxicity mechanism of this alga’s ichthyotoxic effects. Three theories have emerged amid debates and controversies. The first is the presence of mucus or other polysaccharides that resemble lectins, which may cover fish gills, asphyxiating them [84,85,86]. It is, however, unknown whether this mucus represents a fish’s defensive mechanism or an exfoliated material from the algal surface. The creation of organic toxins is the subject of the second theory, where these HABs have been shown to possess a potential neurotoxin, both in situ and in vitro [87,88]. McNabb et al. (2006) [89] used liquid chromatography-mass spectrometry (LCMS) as a diagnostic tool to validate the toxicity of brevetoxin by testing 34 microalgal isolates that are brevetoxin producers. Their results confirmed that only *K. brevis* strains, imported from the USA, produced brevetoxin. All isolates cultured in New Zealand proved non-toxic by LCMS testing, down to very low levels (0.003 pg cell^−1^). The third theory is that this raphidophyte species produces too many reactive oxygen species (ROS), including superoxide (O), hydrogen peroxide (H_2_O_2_), and hydroxy radicals (OH^•^), and that these ROSs are the ichthyotoxic agents [81,90]. Fish gill tissue is likely to be destroyed if ROS concentrations are high enough, which reduces oxygen intake and causes asphyxiation.

### 3.5. Karlodinium

Numerous fish-killing incidents have been linked to the dinoflagellate genus *Karlodinium* in continents throughout the world, including North America, Europe, Southwest Africa, East Asia, and Australia [91,92]. It was previously categorized as *Gymnodinium* or *Gyrodinium* until the now-famous genus was identified. This genus has 15 recognized species as of 2020, of which 6 species have been proven to be toxic (*K. armiger*, *K. azanzae*, *K. conicum*, *K. corsicum*, *K. digitatum*, and *K. veneficum*) [93]. Notably, *K. veneficum* has been linked to innumerable HAB events and mass fish kills [94,95,96]. Karlotoxins, a class of ichthyotoxins that can enhance the ionic susceptibility of cell membranes and ultimately cause osmotic cell expansion and lysis by creating membrane holes with desmethyl sterols, are ideally produced by the two species *K. veneficum* and *K. conicum* [97,98]. Karlotoxins have anti-grazing qualities and are fatal to predators [91,99]. According to different strains of *K. veneficum* as well as different physical properties of water, the toxicity fluctuates and eventually decreases throughout laboratory-based culturing [91,100]. The species *K. armiger* generates karmitoxin, a powerful nanomolar cytotoxin with an -NH_2_ (amino) group, which is distinct from *K. veneficum* and *K. conicum* and was the primary contributor to fish fatalities from *K. armiger* [101,102]. None of these mixoplankton species, *K. veneficum* [103,104], *K. armiger* [105], and *K. azanzae* [106], have been adequately studied in terms of phagotrophy. Although dangerous dinoflagellates and haptophytes frequently exhibit phagotrophy, its relevance with respect to HABs is not yet fully understood [98,107]. The harmful strains of *K. veneficum* and *K. armiger* paralyze their victims with biomolecules before ingesting them [101,108,109]. Of the *Karlodinium* group, only *K. armiger*, *K. conicum*, and *K. veneficum* have been studied. While the former produces karmitoxin, the latter two are known to produce karlotoxins [101]. Both compounds are large polyethers and have identical chemical structures. Chemically, Karmitoxin has a longer C-C backbone (about 60–64 carbon) and a functional amino group than karlotoxins [101]. Karlotoxins are thought to work by creating pores in cell membranes, which cause the target cells to lyse and cause the target animals to die [97]. In fish cell membranes where cholesterol is the predominant sterol, it has been demonstrated that karlotoxins engage with 4-desmethyl sterols to create pores [110]. Given the similar chemical structure of these toxins, it is anticipated that the mode of action of karmitoxin will be comparable to that of karlotoxins. Binzer et al. [102], while analyzing the impact of these algal toxins on fish, inferred that the fish’s life stage is also crucial. Larger fish are much more susceptible than fish larvae, and in a bloom situation, they will perish sooner. They also postulated that additional toxins are perhaps involved or that there may be more than a single pathway for the toxin to exert its effect on fish.

De Salas et al. [111] initially described *K. australe*, another species of *Karlodinium;* however, it was not until 2014 that this species was linked to fish death [112]. *K. australe* has been witnessed in the coastal waters of southeast Asian nations stretching along Malaysia, Singapore, Philippines to the East China Sea, Japan, and Australia [106,113,114]. This species bloomed in the West Johor Strait in February 2014, killing a significant number of fish in the wild and in cages [112]. The West Johor Strait was also filled with blooms of this particular species again the following year, which resulted in the death of hundreds of tons of fish kept in cages [113]. The chemistry of the ichthyotoxic substances produced by the fish-eating *K. veneficum* was not yet understood when they were identified in 2002 [115]. In fish necropsy, Lim et al. [112] noted protruding eyeballs, reddened iris, discolored and peeling skin, reddening at the bottom of fins, and scarred gills with a brownish cast, which are symptoms comparable to those brought on by karlotoxins. *K. australe* was also shown to be a phagotroph capable of eating the cryptophyte *Rhodomonas salina* [111], although the precise mechanism and method of feeding were still unknown [91]. According to Lim et al. (2014) [112], fish deaths seen during these low-cell density blooms indicate that *K. australe* is a very effective fish killer. Based on this survey, it was reported that fish death incidents connected to *K. australe* low-cell density blooms were brought on by the combined effects of toxicity and phagotrophy. In fact, it was reported that micropredation played a more important role than the *K. australe* toxicity on aquatic organisms [98]. This explains why *K. australe* demonstrated fish lethality even at modest blooms at cell densities <2.34 × 10^6^ cells L^−1^, in comparison with *K. veneficum,* which has been shown to cause significant fish kills for the cell density >10^7^ cells L^−1^ [98].

### 3.6. Miscellaneous Reports

Scattered reports of HAB fish kills have also been reported. Although it is unknown, reports of marine animal deaths have been attributed to a powerful exotoxin released by *Pfiesteria* species [116], while other research has shown that *P. shumwayae* killed fish by micropredation rather than exotoxin [117,118]. Those involving other species of HABs, such as *Epinephelus adscensionis* (Osbeck, 1765) and *Melichthys niger,* have been reported repeatedly [119,120]. This impacted thousands of triggerfish and 2200 fry of *Trachinotus ovatus* [121]. *Prymnesium polylepis*, *Phaeocystis* spp., *Chattonella antiqua*, and *Chattonella marina* are known for their exotoxin and/or endotoxin production, resulting in fish kills [122,123,124] or suffocates finfish and shellfish in some cases, by producing copious mucilage [125,126,127,128]. Therefore, fishes suffocate due to excess mucus secretion. *Chaetoceros concavicornis* and *C. convolutus* have extremely sharp cellular extensions that result in capillary hemorrhage, disruption of gas exchange at the gills, excessive mucus production, and suffocation of cultured and wild salmonids [129] and red king crabs and *Paralithodes camtschatica* [130]. *Margalefidinium polykrikoides*, one of the toxic dinoflagellates, was reported to cause fish kills around coastal geographies [131]. According to a study, exposure to *M. polykrikoides* probably impairs gill function and decreases oxygen and food intake as a consequence of lowered metabolic activity of fish [131]. Factors such as phases of life and ontologies of individual species seem to play a role in understanding the toxicity mechanisms of these HABs [131,132]. For instance, *M. polyrikoides* toxins sub-lethal effect was found to cause fish larvae to starve, making them more susceptible to toxin-producing HABs than fed conspecifics [133]. Acute toxicity, sublethal behavioral impacts, and perhaps the decreased nutritional value of the water all contribute to the severity of fish kills [131].

Around the world, paralytic shellfish poisoning (PSP) has been linked to the dinoflagellate genus *Alexandrium* [134]. However, it has been demonstrated that a number of species that do not produce PSP toxins are harmful to finfish [135]. There are more than 40 dinoflagellate species in the genus *Alexandrium*, many of which, like *A. tamarense*, *A. andersonii*, *A. minutum*, and *A. catenella*, have been thoroughly established to be poisonous, primarily because of PSTs [135,136]. However, other *Alexandrium* species, including *A. leei* and *A. ostenfeldii*, have been proven to be ichthyotoxic without generating PSP toxins [135].

Fish fatalities can also happen from direct parasitism by harmless marine dinoflagellate algal blooms [137]. They have been identified to lead to mass mortalities in finfish and shellfish. As an illustration, the parasite *Ichthyodinium chabelsrdi* led to widespread sardine mortality in the Mediterranean [138]. *Amyloodinium ocellatum* is a virulent pathogen in estuaries affecting striped bass [139]. Table 2 enlists the marine fishes killed/affected by HABs.

## 4. Mitigation Methods

The procedure for HAB mitigation involves prevention, management, and control. In case of HAB formation, the goal is to decrease its negative consequences by preventing interaction between marine organisms and the bloom while simultaneously working to restore the natural habitat. The currently practiced mitigation strategies can be categorized into physical, chemical, and biological. Some of the methods that have been employed in fields are categorized and discussed below.

### 4.1. Physical

In marine fish farms, aeration (such as bubble curtains and airlift upwelling) is frequently used to mitigate the harmful effects of HABs that cause water hypoxia or anoxia [160]. The goal of airlift upwelling is to replace surface water that contains a lot of microalgae with deep water. The bottom water functions as a dilution mechanism since it often lacks photosynthetic cells. Results are better when employed with perimeter skirts. The *Pseudochattonella* bloom in Chilean salmon facilities in 2016 could not be controlled using this strategy [161]. The traditional techniques of top and bottom strata mixing of water using hydraulic pumps or hypolimnetic removal and flushing using powerful motors, as employed in North America, were not successful owing to huge costs and the logistic constraints of moving heavy equipment [160].

### 4.2. Chemical

Conventionally, clays were employed to flocculate harmful algal cells [162], but Sengco et al. [163] and Seger et al. highlighted the ability of clay minerals to target the adsorption of ichthyotoxins produced by the haptophyte *P. parvum* [164], the dinoflagellates *Margalefidinium polykrikoides* [165], and *K. brevis* [166] as well as freshwater blooms [167,168]. The idea behind using clay to reduce HAB is based on the flocculation abilities of clay to absorb HABs and drag them to bottom waters so they stop growing or die as a consequence of inhibiting the growth factors responsible for their growth [169]. It was found that clay plays a role in impacting HAB cells in terms of growth rate, antioxidant enzyme activity, and photosynthetic rate. Although HAB cells were not eliminated by flocculation, it may be deduced that surface interaction between modified clay and remaining HAB cells could stimulate those cells and produce a lot of reactive oxygen species (ROS), leading to a significant rise in cell SOD (superoxide dismutase) and CAT (catalase) activity. Therefore, the key mechanism behind inhibiting residual cell growth was probably the augmentation of ROS generation caused by functionalized clays. However, factors such as the huge quantity requirement (110–400 tons/km^2^ of clay), logistical feasibility, and heavy deposition load are some of the challenges that need to be considered when opting for a clay-based mitigation strategy [162,170,171].

### 4.3. Biological

The use of biological agents such as bacteria and viruses has become a novel means to control and manage HABs [172]. The inter-kingdom relationship between bacteria and algae is both complex and dynamic in freshwater and marine environments. They operate in a variety of ways [173]. Usually, the interaction between the bacteria and algae takes place around the “phycosphere”, which is a layer enriched with dissolved organic exudate from algae cells [174]. The mode of algicidal interactions can be both direct, where bacteria lyse algae by physical contact, or indirect approaches, where the chemicals from the bacteria induce algae death [174]. Perhaps the most typical result of algicidal interactions is the lysis, or rupture, of the algal cells [175,176]. Typically, bacteria-triggered algal death occurs without the algal cells experiencing any metabolic or physiological effects [177]. For instance, benzoic acid generated by algicidal *Thalassospira* sp. causes cell lysis of *K. mikimotoi* [178]. The likely mechanism of benzoic acid penetration through the cell membrane and resultant acidification of algal cytoplasm was proposed. Some algae can escape or, at the very least, postpone cell death by producing a protective cyst [174]. When *K. brevis* was grown with algicidal bacteria from the *Cytophaga*/*Flavobacterium*/*Bacteroidetes* (CFB) group, Roth et al. [179] observed the existence of cyst-like cells, indicating that algicidal bacteria may promote a transition to the cyst stage in dinoflagellates. However, this characteristic may not be directly linked to the action of certain bacteria; rather, it is more likely to be a dinoflagellate defense mechanism. After being exposed to bacterial algicides, algae frequently produce reactive oxygen species (ROS) like hydrogen peroxide (H_2_O_2_), the superoxide anion (O_2_^−^), singlet oxygen (1O_2_), and the hydroxyl radical (HO) [174]. Oxidative stress brought on by an excess of ROS causes cell damage and even death [180]. Another interesting field of study that is being explored is to use certain classes of bacteria like Proteobacteria (α, β, and γ), Actinobacteria, and Bacilli to degrade HAB toxins such as microcystin [181]. Microcystins are cyclic heptapeptide liver toxins that are predominantly found in freshwater and marine environments and are chemically stable in water even in both extremes of high and low temperature and pH range without sunlight [181]. The standard water treatment methods have failed to entirely eliminate this potentially carcinogenic toxin. The pathway of toxicity caused by this toxin is through inhibition of protein phosphatases (PP1 and PP2A). Consequently, key biological proteins may become hyperphosphorylated and trigger cellular necrosis, apoptosis, and inflammation, eventually resulting in liver bleeding and DNA damage [182,183]. As an alternative, the HcRNAV virus has been employed in the first field trial to control *Heterocapsa circularisquama* bloom [184]. There are reports on the use of seaweeds to prevent (via nutrient competition/allelochemicals) the development of high biomass HABs [185,186]. Rhodophytes are the most potent group that can inhibit HABs [186]. A few other rhodophytes (red algae) are also able to inhibit HABs by raising the competition for nutrients and by secreting allelopathic compounds [187]. For instance, it has been confirmed that some species of *Gracilaria* native to Asian seas limit the formation of numerous HABs by allelopathy [186,188,189]. *Dasysiphonia japonica* grows faster than other red macroalgae, which helps it compete with other types of algae in nutrient-rich estuaries [190]. No research has been conducted on the possible impacts of *D. japonica* on HABs phytoplankton, including *A. anophagefferens* [191]. Another novel method is the use of algal-lysing substance prodigiosin to target *H. akashiwo*’s cellular shape, antioxidant defenses, molecular gene expression, and finally, cause the death of algal cells [192]. A variety of bacteria, such as Serratia, Streptomyces Zooshikella, and Pseudoalteromonas, can create the red pigment prodigiosin (PG) [193]. However, the limitations of using biological agents include evaluation of cost, geographical location, viability, social acceptance, and proven on-field specificity of biological agents with target HAB management techniques [194,195]. It is important that we emphasize here that, despite such mitigation options being readily available, they have never been tested in the field but only under laboratory conditions.

## 5. Current Gaps in Our Knowledge and Direction for Future Perspective

Numerous harmful algal blooms have been identified to rise from eutrophication or nutrient enrichment [196]. The issue of HABs and their effects on fish and marine ecosystems is a powerful reminder of how critically important a healthy, balanced natural environment is to marine development and how aquaculturists have a stake in preventing environmental degradation. The reports are considerably large in number. However, to date, the mechanism behind fish kills from HABs has not been systematically worked out. Doing so will help us to understand the seriousness of the problem and aid in raising awareness, leading to proper caution. The mechanism that leads to fish kills is different for different HABs (Figure 2); this needs to be meticulously evaluated, also. The primary process appears to vary depending on the HAB species, and additional research is required to determine the mechanism of several notorious algae, such as *K. mikimotoi*. The standard method of using isolated toxins and measuring their toxicity for HAB species may not be useful for examining toxicity mechanisms [197,198]. Moreover, many ichthyotoxic toxins produced by dinoflagellates have not yet been fully identified or characterized, such as those produced by *Margalefidinium polykrikoides*, *M. fulvescens*, *Akashiwo sanguinea*, and *Amphidinium carterae* [135].

Increase and decrease of elemental concentration and pH of water is depicted through upper and lower arrow mark respectively in yellow color in Figure 2. Yan et al. recommended a biotechnology approach of comparing gene expression of both algae and test species to unravel the mechanism [198]. Although, in most cases, it is well-established that HABs kill fish, the exact mechanism that varies with respect to each algal species has yet to be systematically and authoritatively worked out. This review highlights the need for mechanistic studies to be conducted in order to obtain a clear perspective on the mechanism behind the fish kills. There are gaps in the current understanding that need to be addressed in order to gain a clear and holistic perspective of the underlying issues. One such gap is the lack of a fundamental understanding of the toxicity mechanism at work in HABs. Figure 3 gives the results of the PubMed search using the search terms ‘harmful algal blooms and fish kills’; as observed from the graph, the search returned a meager 215 hits. Because the devastation is enormous, it is strange to see limited research enthusiasm for HAB-related fish kills. Most of the reports on fish kills by HABs appear as online news reports or blogs; there are a mere 215 publications represented as scientific publications. This is an evident lack, and this review prompts research attention in this direction.

Yñiguez and Ottong [199] used random forest, a machine learning model for the prediction of fish kills by HABs by employing physical data such as temperature, dissolved oxygen, water pH, salinity, and chlorophyll from 2015 to 2017 in Bolinao-Anda, Philippines. Their model suggested that higher salinity and temperatures increased the likelihood of fish deaths, but lower salinity and higher chlorophyll levels increased the likelihood of hazardous blooms. Another recent study involving a satellite system in the SW Iberian Peninsula [200] was found to be promising for control of HAB.

More such simulation studies are required in this direction. With various sophisticated remote sensing technologies becoming available, prediction models encompassing earth- and satellite-based observations in combination with computer models and regular field sampling can help identify HABs in their infancy so that protective measures and appropriate steps can be taken. The fact that we emphasize here is that the application of advanced technologies in the detection, prevention, and mitigation of HABs is far from implementation. Proper utilization of the available resources could help in addressing HABs appropriately. Most of the mitigation methods have not been attempted on-site but have been confined to laboratory-scale experiments; this is another gap that needs to be addressed. Despite the laboratory-scale success reported using biological mitigation, the question is why this technology has not been implemented. Our PubMed search using the search terms ‘harmful algal blooms’ displayed 2537 hits, yet the search on HAB mitigation received only 105 hits (Figure 4A,B). This indicates the low response from mitigation research groups for voluminous reports on HABs; this review urges accelerated research and implementation of mitigation methods.

There are ample studies that report HAB-based toxicity in fish, but nothing much has been aimed at neutralizing these HAB toxins. Algal blooms, to a certain extent, are inevitable; hence, efforts to neutralize or degrade the toxins they release need to be devised. Microorganism-mediated degradation of HAB toxins, nanomaterial-mediated sorption of HAB toxins, and nanoclay/nanosponge-mediated nanosorption of HAB toxins are a few futuristic directions that may prove beneficial. Additionally, methods to inhibit the proliferation of HABs could also prove useful.

Rules and regulations play a vital role, and the Centre for the Environment, Fisheries and Aquaculture Science (CEFAS) carefully monitors commercial fisheries for signs of toxins. The issue is that most of these regulatory bodies focus on the prevalent, well-reported contaminants and ignore the ones that are infamous for their toxicity. This is yet another gap identified during the course of this review preparation. This review emphasizes that stringent regulations to monitor HABs are lacking and need to be reinforced.

## 6. Conclusions

In this study, the predominant HABs involved in marine fish kills are reported, the various toxins that are responsible, and the speculated toxicity mechanisms behind the fish kills, as well as the mitigation measures available, are discussed. The review highlights the lapses in the implementation of the biological mitigation methods as well as the engagement of nanosorption-based nanotechnological advancements for the resolution of HAB fish kills. The gaps in the proper understanding of the toxicity mechanisms of the various HAB toxins and the lack of involving satellite and remote sensing techniques and machine learning methods for early detection and resolution of HAB issues are projected. Lack of awareness of the new generation of resources is pointed out, and the need to exploit these is prompted.

## Figures and Tables

**Figure 1 plants-12-03936-f001:**
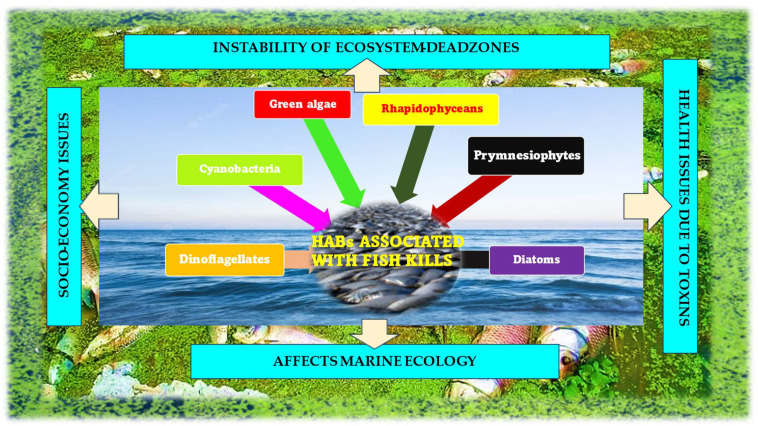
Snapshot of the various algal members that contribute to HABs and their classified impacts of HAB.

**Figure 2 plants-12-03936-f002:**
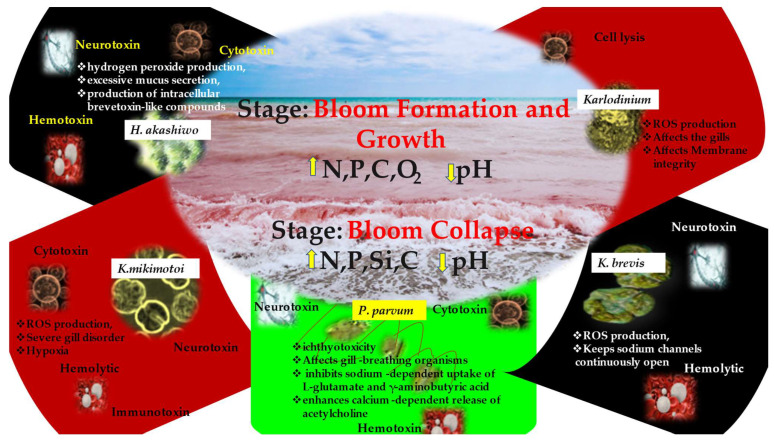
The various mechanisms of HABs involved in fish kills. The up arrows signify ‘increased’ and the down arrow signify ‘decreased’.

**Figure 3 plants-12-03936-f003:**
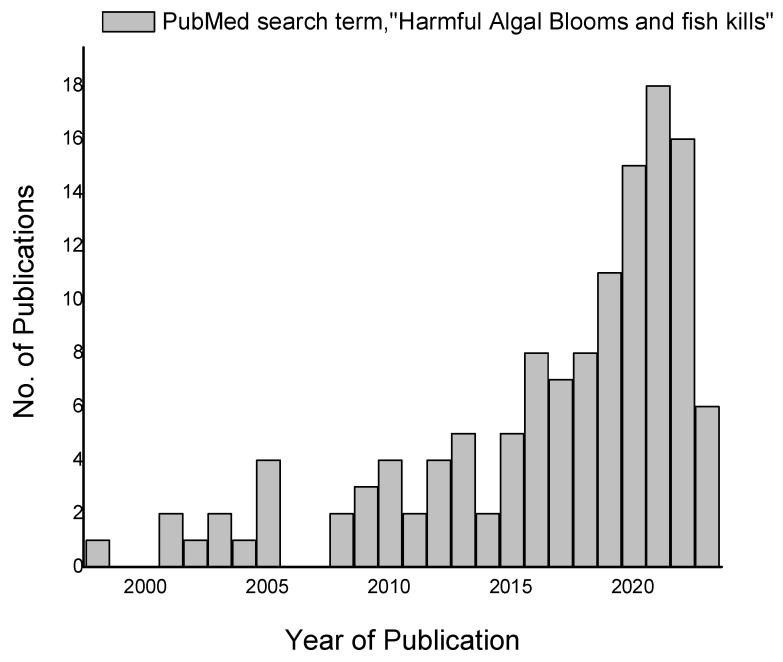
HAB fish kills PubMed search results.

**Figure 4 plants-12-03936-f004:**
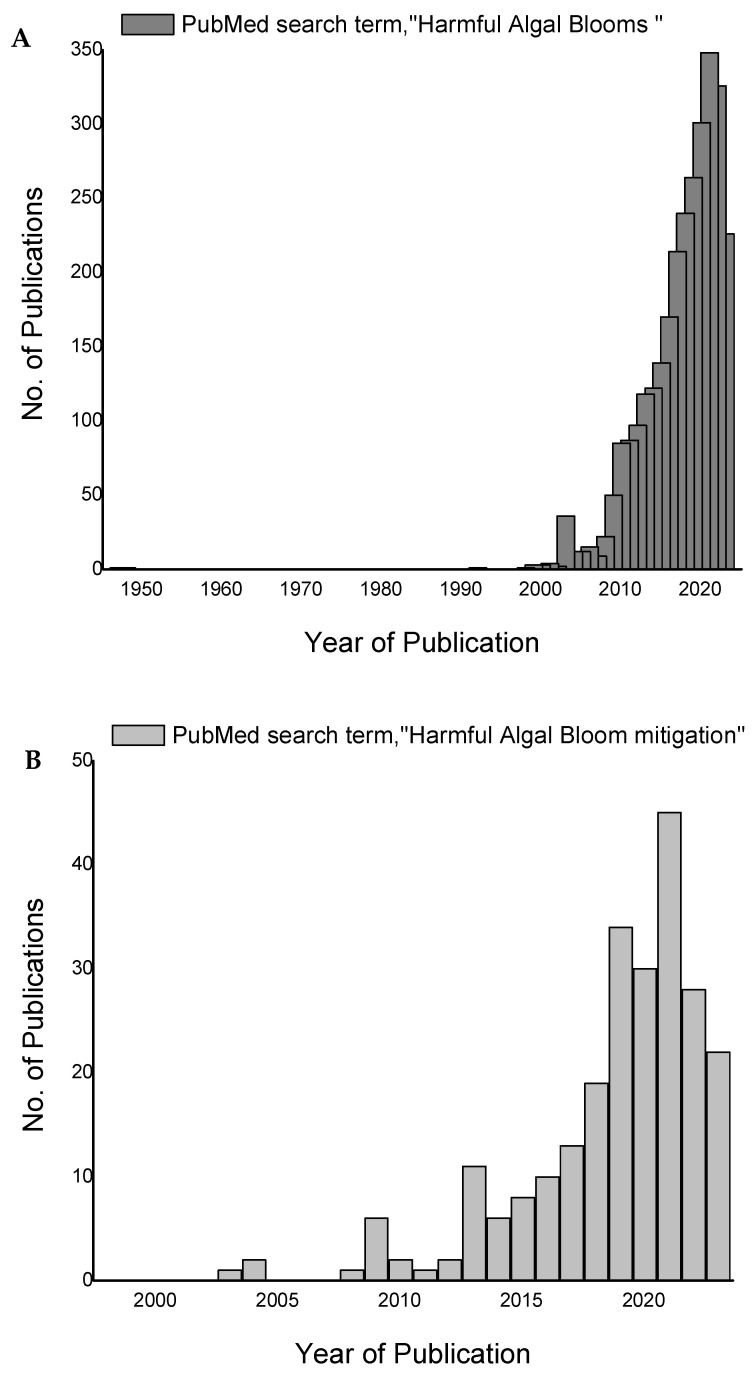
Results of the PubMed search using the search terms: (**A**) ‘harmful algal blooms’ and (**B**) ‘harmful algal bloom mitigation’.

**Table 1 plants-12-03936-t001:** Consolidated list of the predominant HAB events recorded since 2018.

Species	Location	Start Date	End Date	Affected Species Due to Bloom Events
*Karenia mikimotoi* (34,400 cells/L)	Saga, Imari Sea, Northern Kyushu	12 June 2018	11 September 2018	NA
*Karenia mikimotoi* (31,400 cells/L)	Kuzyuukutou, Kusudomari, Western Kyushu	25 June 2018	1 Juky 2018	NA
*Karenia mikimotoi* (1600 cells/L)	Arounndo nobeokasi kitauramati, Eastern Kyushu	11 July 2018	25 July 2018	NA
*Karenia mikimotoi* (84,000,000 cells/L)	Uwajima Bay, Bungo Channel	14 June 2018	21 August 2018	Aquaculture Fish
*Karenia mikimotoi* (22,000,000 cells/L)	Bisan Seto and Hiuchi-nada, Seto Inland Sea	1 August 2018	14 August 2018	Natural Fish, Aquaculture Fish
*Karenia mikimotoi* (66,000,000 cells/L)	Imari Bay, Coast of Saga Prefecture,	11 June 2019	20 August 2019	Aquaculture Fish, Shellfish
*Karenia mikimotoi* (210,500,000 cells/L)	Imari Bay, Coast of Nagasaki Prefecture	11 June 2019	7 August 2019	Aquaculture Fish
*Karenia mikimotoi* (1,250,000 cells/L) (with other blooms)	Yatsushiro Sea, Nagashima~Fukunoura	25 June 2019	6 July 2019	Aquaculture Fish
*Karenia mikimotoi* (37,000,000 cells/L)	Saeki Bay, Bungo Channel	11 June 2018	16 August 2018	Natural Fish, Aquaculture Fish, Shellfish
*Heterosigma akashiwo* (2057 cells/L)-*causative*, *Karenia mikimotoi* (131 cells/L)-*co-occurring*	Wakayama, Kiisuidou, Japan	6 August 2018	11 August 2018	NA
*Karenia brevis* (250,000 cells/L)	Coastal Alabama, USA	5 November 2018	3 December 2018	Natural Fish, Shellfish
*Karenia brevis* (11,000,000 cells/L)	Florida east coast	29 September 2018	18 November 2018	Natural Fish, Humans
*Karenia brevis* (90,000,000 cells/L)	Florida west coast	1 January 2018	31 December 2018	Natural Fish, Birds, Aquatic Mammals, Humans
*Karenia brevis* (340,000 cells/L)	Texas Gulf Coast	11 September 2018	8 October 2018	Natural Fish, Humans
*Karenia brevis* (77,295,560 cells/L)	Southwest Florida (Sarasota, Charlotte, Lee, Collier, and Monroe counties), west coast of Florida	30 September 2019	7 December 2019	Natural Fish, Birds, Aquatic Mammals, Humans
*Karenia brevis* (38,000 cells/L)	Fort Morgan Beach, USA	12 October 2021	10 November 2021	Shellfish, Humans
*Karenia brevis* (1,000,000 cells/L)	Tampa Bay, USA	1 January 2021	1 July 2021	Natural Fish, Shellfish, Aquatic Mammals, Humans
*Karenia brevis* (388,400,000 cells/L)	Gulf of Mexico, southwest Florida	17 October 2022	Bloom is ongoing as of 4 May 2023	Planktonic life, Natural Fish, Birds, Other Terrestrial, Shellfish, Aquatic Mammals
*Heterosigma akashiwo*-co-occuring species	The eastern sea area of Liaodong Bay, Liaoning Province	20 July 2021	22 July 2021	NA

NA—Not Available. All data are gathered from the Harmful Algae Event database, a component of the Harmful Algae Information System within the Intergovernmental Oceanographic Commission (IOC) of UNESCO—http://haedat.iode.org/index.php, accessed on 28 September 2023.

**Table 2 plants-12-03936-t002:** The predominant HABs and their related toxins affecting marine fish/aquatic mammals. * Cetaceans.

Species	Toxin Type	Fin-Fish Species Killed/Affected	References
*K. mikimotoi*	Ichthyotoxic—contact sensitive	Parore, Flounder, Yellow-eyed Mullet, Eel, Goby and spotty (New Zealand), Atlantic Salmon, Sardine, Liza Macrolepis, Sobaity Seabream, Dover sole, Dogfish, Turbot, Sea trout, Rainbow trout, Cod, Coalfish, Black goby, Sprat, *Seriola* sp., Pagrosomus major	[56,57,62,140,141,142,143,144,145]
*Prymnesium parvum*	Ichthyotoxin, cytotoxin, and hemolysin, collectively known as “Prymnesins” [146,147,148]	European plaice, Salmon, Yellow tail, American gizzard shad, African Tilapia, Atlantic Salmon, Rainbow trout	[149,150,151]
*Karenia brevis*	Brevetoxins (neurotoxin)	Ladyfish, Mullet, Pigfish, Spotted Seatrout, Sheepshead, Snook, Rough-toothed Dolphin *, Manatees *, Prey fish, Sharks, Rays	[71,152,153]
*Heterosigma akashiwo*	Ichthyotoxin, neurotoxin, ROS production	Atlantic salmon, Coho salmon, Chinook salmon, Sockeye salmon, Chum salmon, Forage fish	[154]
*Karlodinium* (*K. veneficum, K. conicum*) *K. armiger*	Karlotoxins, ichthyotoxic, hemolytic, cytotoxic, algicidalKarmitoxin, ichthyotoxin	Black bream, Perth herring, estuary cod (*Epinephelus coioides*), *Lates calcarifer*, Paddletail snapper, Four-finger threadfin, *Oryzia melastigma*, Sheepshead minnowLarvae of sheepshead minnow, Rainbow trout, Zebra fish (laboratory study)	[101,102,112,155,156,157]
*Pfiesteria*	Endotoxin	Atlantic Menhaden, Sheepshead minnow larvae	[158,159]

## Data Availability

No new data were created or analyzed in this study. Data sharing is not applicable to this article.

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
