# Peer review of "Review of Harmful Algal Blooms (HABs) Causing Marine Fish Kills: Toxicity and Mitigation"

_plants, 2023, doi:10.3390/plants12233936_

Round 1
Reviewer 1 Report (Previous Reviewer 3)
Comments and Suggestions for Authors
While some previous mistakes have been corrected, this resubmission still requires very significant work before it is acceptable for publication. Unlike the well-studied shellfish toxin producing harmful algal blooms (HABs), fish-killing HABs have been poorly studied (partly because of poor cooperation from the secretive finfish farm industry, partly because of chemical challenges of characterisation of ichthyotoxins dissolved in seawater in low concentrations). While an up to date review on fish killing HABs would be highly valuable, these authors are not experts in HABs and there are numerous misleading/incorrect interpretations of the incomplete and contradictory literature. The manuscript has been sloppily prepared and I include a marked highlighted version of where corrections are needed. eg. All species and genus names need to be in italics (also in References) , with the species name not in capitals, eg line 2073 Nodularia spumigena, not Nodularia Spumigena; When abbreviating Karenia to K. brevis, there should be a space after K, not K.brevis etc.
line 62. I am not clear what the "bright side of this interaction" refers to? biotech potential of ichthyotoxins?, but we don't even know the chemical structures of most of them, and we can only produce them in very low concentrations. The section 5. on potential industrial application of HAB toxins is highly premature. yes for BTX (an endotoxin that you can extract from bulk algal cultures) but not for the fish killing toxins. We cannot yet produce prymnesins, karlotoxins, gymnocin, etc in commercial quantities
line 80., 1652 "HABs are the result of excessive nutrients". This is misleading and only correct for high biomass HABs (cyanobacteria, Phaeocystis, Noctiluca) , not for all HABs; some can kill fish at extremely low concentrations. .See Davidson, K., Gowen, R.J., Harrison, P.J., Fleming, L.E., Hoagland, P., Moschonas, G. 2014. Anthropogenic nutrients and harmful algae in coastal waters. Journal of Environmental Management. 146: 206-216
194. not all belong to algal "family" (wrong word); what do you refer to?cyanobacteria
Line 521. The focus of this review is on fish-killing algae. The long introduction on HABs (foam, scum), dead zones, shellfish poisoning etc is not relevant. Reduce to a few paragraphs and start at line 521. [note Domoic Acid poisoning and Amnesic Shellfish Poisoning are the same thing; line 286; you left out PSP here}
Line 525. "Neurotoxins are particularly prevalent". This is a major mistake. Prymnesins, Karlotoxins etc damage fish gill membranes (sometimes called lysins) but are NOT neurotoxins and have no human health impacts. That's why the erroneous early claim (non-specific ELISA and HPLC without proper standards) of brevetoxin from raphidophytes (which I raised previously; but the only correction has been to insert "disputed" in line 1078) needs more comment. Carefully read the LCMS verdict (looking for diagnostic ions) by : McNabb P, Rhodes L, Adamson J, Holland P (2006) Brevetoxin — an elusive toxin in New Zealand waters, African Journal of Marine Science 28, 375-377.
The section 4 (much underdone) on mitigation applies to all HABs not just fish-killing HABs
1611. "not many are in continuous use" ; biological control has never been used in a field situation, only in the lab
1657. The use of seaweeds to reduce HABs has been primarily as nutrient competition/allelochemicals against high biomass HABs
Fig.4. Panel 2 has the direction of the time scale 2022 to 2001 reversed (again this is sloppy!). The 4 panels are repetitive. Freshwater fish kills can be deleted because that is not covered here. A geographic breakdown where those papers come from many been useful?
Line 1810. "contamination"; "detection of allied toxins in fishes" ; "HAB polluted fishes". This is dangerously misleading. With the exception of brevetoxin (only produced by Karenia brevis not by any other HAB), none of the ichthyotoxins are harmful to humans; Fish are killed because of gill damage, and subsequent suffocation, but fish flesh is never contaminated by dangerous (to humans) toxins.
722, gymnocin responsible. This is wrong. It has never been proven that gymnocin can occur in seawater at sufficient concentrations to kill fish. We don't even know how to measure gymnocin in seawater.
Line 983. See Dorantes-Aranda. Brevetoxins do not damage fish gills!
1086. Leadbeater & Dodge are not the authors of the genus Karlodinium
Throughout line 1217 ,912. The terminology "excretes" , "exotoxin and endotoxin" line 1215 etc are carelessly used. For Karlodinium yes we know that karlotoxins can be excreted, but for Chrysochromulina and Karenia brevis this is not at all clear, which means that algal cells need to lyse for ichthyotoxicity to occur. The authors overlooked the commonality that most fish killing algae are very fragile and can lyse even upon impact on the gills of fish
1217 "distant parasitism" , this a wrong analogy!
1214. Chrysochromulina polylepis is now called Prymnesium polylepis
1359. Do not abbreviated Alexandrium as Al. but A.
1349. Cochlodinium polykrikoides is now Margalefidinium polykrikoides (correct on line 1668, Pheopolykrikos is not ichthyotoxic)
1402. "This incident proved". HAB=harmful algal bloom, per definition has a harmful impact on human society; there is nothing to prove here.
1520. Clay is widely used in Korea, not Japan, but of increasing interest in China
1693. "ideally not much enthusiasm evident"? sloppy language; what do you mean to say
1493. The purpose of clay flocculation in the past has been primarily for flocculation (ie. attach, not absorb to HABs) but new research is focusing on mapping up ichthyotoxins; Seger, A and Hallegraeff, G. 2022.Harmful Algae 111, 102151 https://doi.org/10.1016/j.hal.2021.102151.
The objections against clay do not relate to "pollution" but smothering of benthic invertebrates, see Shumway, S.E., D.M. Frank, L.M Ewart, and J.E. Ward. 2003. Effect of yellow loess on clearance rate in seven species of benthic, filter-feeding invertebrates. Aquaculture Research 34:1391–1402.

Very numerous improvements to English grammar are needed.
Title. Reviewing=Review of HABs on =causing marine fish kills etc.
Fig.1.Socio-economic impacts. This figure applies to all HABs
Fig.2 Raphidophyceans +add Dictyochophytes (incl Pseudochattonella). Green algae should not be there. This figure is rather meaningless
Fig.3 K.brevis. brevetoxin does not block voltage gated sodium channels but keeps them continuously opened
551. fish have "bare skin" is poor language
806. Pacific salmon
813. defaulty? what do you want to say
1091. what is an omni species?
1309. costed=cost
1404.practised (verb)
1653 fishes=fish
1659. till date=until to date
1813. undermine?
1909. elaborately?
Author Response
We would like to thank our Editor for having given us the revision opportunity. We thank the reviewers for their extensive suggestions. Thank you for your generous efforts. We have now responded to the queries in line and made necessary modifications to the manuscript. Thank you.
Reviewer 1
While some previous mistakes have been corrected, this resubmission still requires very significant work before it is acceptable for publication. Unlike the well-studied shellfish toxin producing harmful algal blooms (HABs), fish-killing HABs have been poorly studied (partly because of poor cooperation from the secretive finfish farm industry, partly because of chemical challenges of characterisation of ichthyotoxins dissolved in seawater in low concentrations). While an up to date review on fish killing HABs would be highly valuable, these authors are not experts in HABs and there are numerous misleading/incorrect interpretations of the incomplete and contradictory literature. The manuscript has been sloppily prepared and I include a marked highlighted version of where corrections are needed. eg. All species and genus names need to be in italics (also in References) , with the species name not in capitals, eg line 2073 Nodularia spumigena, not Nodularia spumigena; When abbreviating Karenia to K. brevis, there should be a space after K, not K.brevis etc.
Ans. We thank you for your patience and most valuable suggestions. We have revised the manuscript. Also, we have made sure the scientific nomenclature has been kept. Thank you.
line 62. I am not clear what the "bright side of this interaction" refers to? biotech potential of ichthyotoxins?, but we don't even know the chemical structures of most of them, and we can only produce them in very low concentrations. The section 5. on potential industrial application of HAB toxins is highly premature. yes for BTX (an endotoxin that you can extract from bulk algal cultures) but not for the fish killing toxins. We cannot yet produce prymnesins, karlotoxins, gymnocin, etc in commercial quantities
Ans. One of the previous reviewers had asked us to add a section on the industrial use of HABs, since you were not comfortable, we have removed this section in the revised version. Thank you.
line 80., 1652 "HABs are the result of excessive nutrients". This is misleading and only correct for high biomass HABs (cyanobacteria, Phaeocystis, Noctiluca) , not for all HABs; some can kill fish at extremely low concentrations. .See Davidson, K., Gowen, R.J., Harrison, P.J., Fleming, L.E., Hoagland, P., Moschonas, G. 2014. Anthropogenic nutrients and harmful algae in coastal waters. Journal of Environmental Management. 146: 206-216
Ans. We modified these sentences. Thank you.
- not all belong to algal "family" (wrong word); what do you refer to?cyanobacteria
Ans. Modified
Line 521. The focus of this review is on fish-killing algae. The long introduction on HABs (foam, scum), dead zones, shellfish poisoning etc is not relevant. Reduce to a few paragraphs and start at line 521. [note Domoic Acid poisoning and Amnesic Shellfish Poisoning are the same thing; line 286; you left out PSP here}
Ans. Yes, Modified. Thank you.
Line 525. "Neurotoxins are particularly prevalent". This is a major mistake. Prymnesins, Karlotoxins etc damage fish gill membranes (sometimes called lysins) but are NOT neurotoxins and have no human health impacts. That's why the erroneous early claim (non-specific ELISA and HPLC without proper standards) of brevetoxin from raphidophytes (which I raised previously; but the only correction has been to insert "disputed" in line 1078) needs more comment. Carefully read the LCMS verdict (looking for diagnostic ions) by : McNabb P, Rhodes L, Adamson J, Holland P (2006) Brevetoxin — an elusive toxin in New Zealand waters, African Journal of Marine Science 28, 375-377.
Ans. We have checked on this and done the needful. Thank you.
The section 4 (much underdone) on mitigation applies to all HABs not just fish-killing HABs
Ans. Yes, we had kept this section general not just specific to fish killing HABs, since as such there wasn’t much information even on the general context. We have now revised and updated this section too. Thank you.
- "not many are in continuous use" ; biological control has never been used in a field situation, only in the lab
Ans. Rewritten the sentence. Thank you.
- The use of seaweeds to reduce HABs has been primarily as nutrient competition/allelochemicals against high biomass HABs
Ans. Modified
Fig.4. Panel 2 has the direction of the time scale 2022 to 2001 reversed (again this is sloppy!). The 4 panels are repetitive. Freshwater fish kills can be deleted because that is not covered here. A geographic breakdown where those papers come from many been useful?
Ans. Figure revised.
Line 1810. "contamination"; "detection of allied toxins in fishes" ; "HAB polluted fishes". This is dangerously misleading. With the exception of brevetoxin (only produced by Karenia brevis not by any other HAB), none of the ichthyotoxins are harmful to humans; Fish are killed because of gill damage, and subsequent suffocation, but fish flesh is never contaminated by dangerous (to humans) toxins.
Ans. Corrected. Rephrased.
722, gymnocin responsible. This is wrong. It has never been proven that gymnocin can occur in seawater at sufficient concentrations to kill fish. We don't even know how to measure gymnocin in seawater.
Ans. Removed
Line 983. See Dorantes-Aranda. Brevetoxins do not damage fish gills!
Ans. Repharased, thank you.
- Leadbeater & Dodge are not the authors of the genus Karlodinium
Ans. removed
Throughout line 1217 ,912. The terminology "excretes" , "exotoxin and endotoxin" line 1215 etc are carelessly used. For Karlodinium yes we know that karlotoxins can be excreted, but for Chrysochromulina and Karenia brevis this is not at all clear, which means that algal cells need to lyse for ichthyotoxicity to occur. The authors overlooked the commonality that most fish killing algae are very fragile and can lyse even upon impact on the gills of fish
Ans. Revised.
1217 "distant parasitism" , this a wrong analogy!
Ans. Removed
- Chrysochromulina polylepis is now called Prymnesium polylepis
Ans. Changed
- Do not abbreviated Alexandrium as Al. but A.
Ans. Corrected. Thank you.
- Cochlodinium polykrikoides is now Margalefidinium polykrikoides (correct on line 1668, Pheopolykrikos is not ichthyotoxic)
Ans. Corrected
- "This incident proved". HAB=harmful algal bloom, per definition has a harmful impact on human society; there is nothing to prove here.
Ans. Rephrased
- Clay is widely used in Korea, not Japan, but of increasing interest in China
Ans. Corrected
- "ideally not much enthusiasm evident"? sloppy language; what do you mean to say
Ans. rephrased
- The purpose of clay flocculation in the past has been primarily for flocculation (ie. attach, not absorb to HABs) but new research is focusing on mapping up ichthyotoxins; Seger, A and Hallegraeff, G. 2022.Harmful Algae 111, 102151 https://doi.org/10.1016/j.hal.2021.102151.
Ans. Elaborated and rephrased.
The objections against clay do not relate to "pollution" but smothering of benthic invertebrates, see Shumway, S.E., D.M. Frank, L.M Ewart, and J.E. Ward. 2003. Effect of yellow loess on clearance rate in seven species of benthic, filter-feeding invertebrates. Aquaculture Research 34:1391–1402.
Ans. Elaborated based on the reference. Thank you.
Very numerous improvements to English grammar are needed.
Title. Reviewing=Review of HABs on =causing marine fish kills etc.
Ans. Revised. Thank you.
Fig.1.Socio-economic impacts. This figure applies to all HABs
Ans. Clarified
Fig.2 Raphidophyceans +add Dictyochophytes (incl Pseudochattonella). Green algae should not be there. This figure is rather meaningless
Ans. Revised
Fig.3 K.brevis. brevetoxin does not block voltage gated sodium channels but keeps them continuously opened
Ans. corrected
- fish have "bare skin" is poor language
Ans. Sorry, rectified
- Pacific salmon
Ans. Corrected
- defaulty? what do you want to say
Ans. Rephrased
- what is an omni species?
Ans. Clarified
- costed=cost
Ans. Corrected
1404.practised (verb)
Ans. Rectified
1653 fishes=fish
Ans. Revised
- till date=until to date
Ans. revised
- undermine?
Ans. revised
- elaborately?
Ans. Revised, thank you. We would like to appreciate all your very valuable suggestions. This has certainly improved the quality of our manuscript, thank you very much, appreciate the effort and time.
Reviewer 2 Report (New Reviewer)
Comments and Suggestions for Authors
GENERAL COMMENTS
Literature is plenty of reviews on HAB, but those focusing only on fish mortality are not so much. For this reason, this review could be interesting.
However, I found this ms an awkward try to get together what authors found about this argument. The English style is not good, the discourse is not fluid, often in the text one passes from one topic to another, and then returns to the previous one a few lines below, in some points it is really difficult to follow the flow. Moreover, my main concern about this ms is the lack of conclusion about the argument. This is evident reading the abstract and the conclusion.
In the Abstract, it is just written that “this review summarizes the reports on various HABs, which have been able to affect and bring about marine fish kills. The predominant HABs, their toxins, and their effects on fishes spread across various parts of the globe have been discussed” this is right obvious as it is a review about this argument, instead give the names of the most important microalgal species, the names of involved toxins ...
And going ahead:
“From what is available, the mechanism of HABs toxicity in fish, and the challenges, and limitations facing the development and implementation of mitigation measures have been discussed and presented. The gaps in the existing knowledge of HABs and their control have been addressed”.
Honestly, I do not mind that all these points have been addressed in the ms because it is obvious, I would like to have some anticipations of them here.
And :
“Recommendations and suggestions to improvise the gaps in the current knowledge have been highlighted. The totality of the impact of algal blooms has been viewed taking into consideration the dark and bright side of this interaction.” Again, supply the recommendation and suggestion here.
Regarding the conclusion section, no conclusions are written down at all:
“The current article reviews the effect of HABs on marine fishes, and published reports on the different species of predominant HABs have been discussed. The seriousness of the issue and its impact on the environment and on marine life and eventually on humans, has been elaborately presented. The gaps in the existing knowledge and research in this direction have been discussed.”
Here we have the same style of the abstract. instead, tell me the different species of predominant HAB, list the main issues and impacts, reveal the gaps in the knowledge and research…
Finally, figures 1-3 do not make any sense to me.
I do not recommend the publication of this ms. A lot of work is still needed to make acceptable this review.
SPECIFIC COMMENTS
L13 “Although infamous and less addressed” I do not understand, it seems contradictory
L36. “ produce’s toxins” ???
L42-43. “HABs secrete a variety of toxins that include: anatoxins, microcystins, neurotoxins, saxitoxin, cylindrospermopsin and nodularin” not only these, there are several others from marine microalgae (here you have listed only cyanotoxins).
L53-54. Figure 1 does not make any sense to me. Please explain it. In particular I cannot understend the arrows: it seams that HAB impacted the nutrient pollution, the bio-invasion and the CLIMATE CHANGE, then the socio-economy issues lead, by means of the climate change, health issue due to the toxins… I really do not understand.
L62-63, “As a fact, not all species trigger HABs, and those that cause blooms do not only belong to the algal family, according to IOC-UNESCO data” what do you mean? HAB (harmful algal bloom) are caused by ALGAE…
L100. “Domoic Acid Poisoning and Amnesic Shellfish Poisoning” please explain the differences among these two poisonings.
L101. “Diarrheic Shellfish Poisoning (DSP)” if I am not wrong, none of the above mentioned toxins are involved in DSP.
L130. Figure 2. Are you really sure that this figure is necessary? This gross figure can be easily substituted with the following sentence: the algal groups that can produce HAB associated to fish death are: diatoms, dinoflagellates, cyanobacteria, ….
L344. What do you mean with “omni species”?
L390. What do you mean with “hypo cell densities”?
L396. “it explain's why” ???
L484-485. “According to prior field experience, a clearance efficacy of 70%–80% would be adequate to control HAB” which kind of HAB? Please give the species name(s)
L489-490. “As a result, additional laboratory studies on the effects of modified clays on HAB organisms were studied” which kind of HAB? Please give the species name(s)
L527-535. I do not find appropriate this part, you are out of topic (here you are discussing the biological mitigation methods)
L548-549. Maybe you should move this part in te part where you discuss the biological mitigation methods
L582. In this panel of figure 4, years are listed in wrong way.
Comments on the Quality of English LanguageThe English style is not good, the discourse is not fluid, often in the text one passes from one topic to another, and then returns to the previous one a few lines below, in some points it is really difficult to follow the flow.
Author Response
GENERAL COMMENTS
Literature is plenty of reviews on HAB, but those focusing only on fish mortality are not so much. For this reason, this review could be interesting.
Ans. Thank you. We have now revised and kept the flow and concurrence in the manuscript. Thank you for your efforts and suggestions.
However, I found this ms an awkward try to get together what authors found about this argument. The English style is not good, the discourse is not fluid, often in the text one passes from one topic to another, and then returns to the previous one a few lines below, in some points it is really difficult to follow the flow. Moreover, my main concern about this ms is the lack of conclusion about the argument. This is evident reading the abstract and the conclusion.
Ans. Revised based on all these points. Thank you for the valuable comments. They helped us in orienting our revision.
In the Abstract, it is just written that “this review summarizes the reports on various HABs, which have been able to affect and bring about marine fish kills. The predominant HABs, their toxins, and their effects on fishes spread across various parts of the globe have been discussed” this is right obvious as it is a review about this argument, instead give the names of the most important microalgal species, the names of involved toxins ...
Ans. Rewritten abstract
And going ahead:
“From what is available, the mechanism of HABs toxicity in fish, and the challenges, and limitations facing the development and implementation of mitigation measures have been discussed and presented. The gaps in the existing knowledge of HABs and their control have been addressed”. Honestly, I do not mind that all these points have been addressed in the ms because it is obvious, I would like to have some anticipations of them here.
Ans. We have added this information.
And :
“Recommendations and suggestions to improvise the gaps in the current knowledge have been highlighted. The totality of the impact of algal blooms has been viewed taking into consideration the dark and bright side of this interaction.” Again, supply the recommendation and suggestion here.
Ans. Section 4 we have suggested our recommendations.
Regarding the conclusion section, no conclusions are written down at all:
“The current article reviews the effect of HABs on marine fishes, and published reports on the different species of predominant HABs have been discussed. The seriousness of the issue and its impact on the environment and on marine life and eventually on humans, has been elaborately presented. The gaps in the existing knowledge and research in this direction have been discussed.” Here we have the same style of the abstract. instead, tell me the different species of predominant HAB, list the main issues and impacts, reveal the gaps in the knowledge and research…
Ans. We have rewritten the conclusion section. Thank you.
Finally, figures 1-3 do not make any sense to me.
Ans. We have revised these figures.
I do not recommend the publication of this ms. A lot of work is still needed to make acceptable this review.
Ans. We have made sincere efforts to revise the manuscript based on all the comments to our best ability. We hope you will reconsider your decision. Thank you in advance.
SPECIFIC COMMENTS
L13 “Although infamous and less addressed” I do not understand, it seems contradictory
Ans. Revised
L36. “ produce’s toxins” ???
Ans. Revised
L42-43. “HABs secrete a variety of toxins that include: anatoxins, microcystins, neurotoxins, saxitoxin, cylindrospermopsin and nodularin” not only these, there are several others from marine microalgae (here you have listed only cyanotoxins).
Ans. Rephrased to clarify.
L53-54. Figure 1 does not make any sense to me. Please explain it. In particular I cannot understend the arrows: it seams that HAB impacted the nutrient pollution, the bio-invasion and the CLIMATE CHANGE, then the socio-economy issues lead, by means of the climate change, health issue due to the toxins… I really do not understand.
Ans. Revised this figure now.
L62-63, “As a fact, not all species trigger HABs, and those that cause blooms do not only belong to the algal family, according to IOC-UNESCO data” what do you mean? HAB (harmful algal bloom) are caused by ALGAE…
Ans. Rephrased
L100. “Domoic Acid Poisoning and Amnesic Shellfish Poisoning” please explain the differences among these two poisonings.
Ans. They are both the same, hence removed one. Thank you for pointing out.
L101. “Diarrheic Shellfish Poisoning (DSP)” if I am not wrong, none of the above mentioned toxins are involved in DSP.
Ans. Yes you are right, we added the respective toxin. We have checked and corrected.
L130. Figure 2. Are you really sure that this figure is necessary? This gross figure can be easily substituted with the following sentence: the algal groups that can produce HAB associated to fish death are: diatoms, dinoflagellates, cyanobacteria, ….
Ans. Combined Fig 1 and 2 as a composite figure.
L344. What do you mean with “omni species”?
Ans. Removed
L390. What do you mean with “hypo cell densities”?
Ans. modified
L396. “it explain's why” ???
Ans. repharased
L484-485. “According to prior field experience, a clearance efficacy of 70%–80% would be adequate to control HAB” which kind of HAB? Please give the species name(s)
Ans. Removed since additional information was not given.
L489-490. “As a result, additional laboratory studies on the effects of modified clays on HAB organisms were studied” which kind of HAB? Please give the species name(s)
Ans. Removed.
L527-535. I do not find appropriate this part, you are out of topic (here you are discussing the biological mitigation methods)
Ans. removed
L548-549. Maybe you should move this part in te part where you discuss the biological mitigation methods
Ans. moved
L582. In this panel of figure 4, years are listed in wrong way.
Ans. Added new figure 4. Thank you for your patience and time. Thank you.
Round 2
Reviewer 1 Report (Previous Reviewer 3)
Comments and Suggestions for Authors
This manuscript has much improved and is suitable for publication. There are still a number of English grammar and format corrections to be fixed up however.
line 34. "Till date"=To date
line 44. biomagnify (one word)
line 95. okadaic acid and ...; something is missing here: ?"ciguatoxin"
line 301,302. Gymnodinium and Gyrodinium in italics
line 304. and; not in italics
line 319. Karlodinium; capital K and in italics
line 351. explains
line 383. Alexandrium; in italics
line 399. practised
line 423. clay plays a role (singular)
line 436. viruses (plural)
line 487. lapses=gaps in our knowledge
line 503. carterae
Check for italics in species names in references (with the species epithet in lower case)
586. Karenia brevis
625, 628 . Prymnesium parvum
634 . Prymnesium
651. Cyprinodon variegatus
652. Dinophysis acuminata
677. Braarud and Heimdal, missing initials
693. G.cf. ; something missing
732. Chattonella marina
766. Acartia tonsa
773. K.armiger
800. Karenia digitata
808. Pfiesteria shumwayae
826. Chrysochromulina
831. Chaetoceros convolutus
850. M. saitlis x M. chrysops
856. Karenia umbella
874. Steno bredanensis
891. Pfiesteria
895. Pseudochattonella
937. wrong font
952. Dasysiphonia japonica
965. Smayda; initials missing.
Comments on the Quality of English Language
English grammar corrections are suggested.
Author Response
This manuscript has much improved and is suitable for publication. There are still a number of English grammar and format corrections to be fixed up however.
Ans. We would like to thank our reviewer for the time and patient inputs and for the encouraging words to appreciate our revision. We have once again revised based on your comments.
line 34. "Till date"=To date
Ans. Modified
line 44. biomagnify (one word)
Ans. Corrected
line 95. okadaic acid and ...; something is missing here: ?"ciguatoxin"
Ans. Corrected
line 301,302. Gymnodinium and Gyrodinium in italics
Ans. Changed
line 304. and; not in italics
Ans. Yes done
line 319. Karlodinium; capital K and in italics
Ans. Corrected
line 351. explains
Ans. Corrected
line 383. Alexandrium; in italics
Ans. Done
line 399. practised
Ans. Done
line 423. clay plays a role (singular)
Ans. corrected
line 436. viruses (plural)
ans. Corrected
line 487. lapses=gaps in our knowledge
Ans. Changed
line 503. carterae
Check for italics in species names in references (with the species epithet in lower case)
Ans. changed
586. Karenia brevis
Ans. changed
625, 628 . Prymnesium parvum
Changed
634 . Prymnesium
Changed
651. Cyprinodon variegatus
Done
652. Dinophysis acuminata
Done
677. Braarud and Heimdal, missing initials
Added
693. G.cf. ; something missing
Done
732. Chattonella marina
Ans. Corrected
766. Acartia tonsa
Corrected
773. K.armiger
Ans. Corrected
800. Karenia digitata
Ans. Corrected
808. Pfiesteria shumwayae
Ans. Changed
826. Chrysochromulina
Ans. Changed
831. Chaetoceros convolutus
Ans. changed
850. M. saitlis x M. chrysops
Ans. Changed
856. Karenia umbella
Ans. changed
874. Steno bredanensis
Ans. Done
891. Pfiesteria
Ans corrected
895. Pseudochattonella
Ans. corrected
937. wrong font
ANs. Corrected
952. Dasysiphonia japonica
Ans. Changed
965. Smayda; initials missing.
Ans. Added
This manuscript is a resubmission of an earlier submission. The following is a list of the peer review reports and author responses from that submission.
Round 1
Reviewer 1 Report
Comments and Suggestions for Authors
This study doesn’t have scientific sound, and as a review paper, it doesn’t contribute or summarise adequately the HABs phenomena. This manuscript has mistakes, e.g. lack of species re-classification names and other flaws that make this work not suitable for publication.
Reviewer 2 Report
Comments and Suggestions for Authors
Harmful algal blooms (HABs) are one of the most serious marine ecological disaster gloablly, consequently, which are associated with marine animal mortality. This review summarized the progress of the toxic activity of harmful algal blooms (HABs) on marine fishes, the topic is instresting, but the organization and writing are not good. I would recommend it to be accepted by this journal after the following revisions.
1. The section "Abstract" is not well prepared, at least it cannot contain the whole content.
2. One serious issue is found throughtout the manuscript, many references are missed, for example, lines 26-35, lines 51-55, lines 85-92......
3. Many informations are out of date, for example, in line 44, Gymnodinium breve has been reclassified as Karenia brevis 20 years ago; Sournia (1995) summarized about 200 types HABs, but now it is 2023. There are many reviews renewed the number.
4. In section "3. Surveying the reports of HABs toxicity on fishes", the authors only reviewed very limited numbers of HABs species, however, many Alexandrium and Margalefidinium species caused serious fish kills, but I cannot see the reviews on these important species.
Reviewer 3 Report
Comments and Suggestions for Authors
This review paper aims to "present a comprehensive overview of up to date reports on HABs affecting marine fish resulting in fish kills" (lines 79-83). This is a worthwhile topic, but unfortunately the ms includes a very large amount of material in the introduction but also discussion (eg. shellfish toxins, freshwater cyanobacterial toxins, industrial application from HAB toxins, Gambierdiscus etc etc) that is completely irrelevant. All of that should come out.
Furthermore, the coverage of this topic is highly incomplete, and this review contains numerous significant mistakes.
Fig.2 includes mistakes (the dinoflagellate picture looks like a euglenoid) but also is grossly incomplete, eg. Prymnesiophytes excludes Prymnesium polylepis and related Chrysochromulina leadbeateri, Dictyochophytes Vicicitus globosa and Pseudochattonella verruculosa, dinoflagellates Karenia, Karlodinium, Takayama. The term red tide is more commonly associated with water discolorations and shellfish toxins not fish kills.
Line 150. It is incorrect that neurotoxins are prevalent in fish kills. Fish Kills are most commonly caused by gill damaging metabolites.
Lines 199-200. The quantitative role of gymnocins and gymnodimines (the latter from Karenia selliformis not K.mikimotoi) remains to be demonstrated.
line 307. The production of brevetoxin by Heterosigma has been disputed
Line 397. Similarly, the quantitative role of the Pfiesteria exotoxin is disputed
Line 403. Nodularia hepatotoxins have not been related to fish kills.
Line 427. the Chilean fish kill was caused by Pseudochattonella not Pfiesteria
I do not support publication of this work in this format, and also query its suitability for this journal

Author Response
This review paper aims to "present a comprehensive overview of up to date reports on HABs affecting marine fish resulting in fish kills" (lines 79-83). This is a worthwhile topic, but unfortunately the ms includes a very large amount of material in the introduction but also discussion (eg. shellfish toxins, freshwater cyanobacterial toxins, industrial application from HAB toxins, Gambierdiscus etc etc) that is completely irrelevant. All of that should come out.
Furthermore, the coverage of this topic is highly incomplete, and this review contains numerous significant mistakes.
Fig.2 includes mistakes (the dinoflagellate picture looks like a euglenoid) but also is grossly incomplete, eg. Prymnesiophytes excludes Prymnesium polylepis and related Chrysochromulina leadbeateri, Dictyochophytes Vicicitus globosa and Pseudochattonella verruculosa, dinoflagellates Karenia, Karlodinium, Takayama. The term red tide is more commonly associated with water discolorations and shellfish toxins not fish kills.
Line 150. It is incorrect that neurotoxins are prevalent in fish kills. Fish Kills are most commonly caused by gill damaging metabolites.
Lines 199-200. The quantitative role of gymnocins and gymnodimines (the latter from Karenia selliformis not K.mikimotoi) remains to be demonstrated.
line 307. The production of brevetoxin by Heterosigma has been disputed
Line 397. Similarly, the quantitative role of the Pfiesteria exotoxin is disputed
Line 403. Nodularia hepatotoxins have not been related to fish kills.
Line 427. the Chilean fish kill was caused by Pseudochattonella not Pfiesteria
I do not support publication of this work in this format, and also query its suitability for this journal